

# A degeneracy bound for homogeneous topological order

**Jeongwan Haah⋆**

Microsoft Quantum, Redmond, Washington, USA

⋆ jwhaah@microsoft.com

## Abstract

We introduce a notion of homogeneous topological order, which is obeyed by most, if not all, known examples of topological order including fracton phases on quantum spins (qudits). The notion is a condition on the ground state subspace, rather than on the Hamiltonian, and demands that given a collection of ball-like regions, any linear transformation on the ground space be realized by an operator that avoids the ball-like regions. We derive a bound on the ground state degeneracy $\mathcal{D}$ for systems with homogeneous topological order on an arbitrary closed Riemannian manifold of dimension $d$, which reads

$$\log \mathcal{D} \le c\mu(L/a)^{d-2}.$$

Here, $L$ is the diameter of the system, $a$ is the lattice spacing, and $c$ is a constant that only depends on the isometry class of the manifold, and $\mu$ is a constant that only depends on the density of degrees of freedom. If $d = 2$, the constant $c$ is the (demi)genus of the space manifold. This bound is saturated up to constants by known examples.

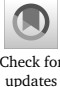
# 1   Introduction

Fracton order refers to perturbatively stable gapped phases of matter that have excitations of restricted mobility [1, 2]. The phases share an important property with conventional topological order that there is no local observable for degenerate ground state subspaces. Beyond this aspect, there is not much that is purely topological in fracton phases: there exist analogs of Wilson loop operators but they only give a many-to-one map into homology groups; continuum field theories have been studied [3–5] but complete data of operators on the ground state subspace still depend on geometric details. Recently [6–8], it is proposed that fracton phases are obtained by stitching together blocks of conventional topological order (anomalous or not), providing a machinery to write a vast number of examples. This construction requires so many algebraic quantities and parameters, including length scales of constituent blocks, that we are motivated to pause and ask what it means for a many-body state to represent a quantum phase of *homogeneous* matter. Translation invariance seems natural but the spatial manifold does not always have a canonical translation group action.

In this article, while we do not directly attempt to answer this question of homogeneity, we propose a general condition for spatial homogeneity of "topological" many-body states, which holds for all much-studied examples to the author's knowledge, yet rules out many situations that are unreasonable from physical perspectives. To demonstrate nontrivial mathematical content of the condition, we prove a sharp bound on the ground state degeneracy.

Recall that one of prominent characteristics of fracton phases is that there are infinitely many superselection sectors. A finitary statement is that a lattice Hamiltonian in a fracton phase on a $d$-torus of linear length $L$ with a fixed finite dimensional degrees of freedom (qudits) per site, the ground state degeneracy $\mathcal{D}$ is given by a diverging function from system sizes $L$ to positive integers. For most relevant constructions, we know that $\log \mathcal{D}$ is a topology-dependent constant in $d = 2$ and $\log \mathcal{D}$ is proportional to $L$ in $d = 3$. In fact, it is absurdly trivial to achieve $\log \mathcal{D} \sim L^{d-2}$ for any spatial dimension $d \geq 2$, by simply stacking any topological ordered state in two dimensions along all but two directions in a $d$-space. The homogeneity condition that we are going to propose implies that this growth rate of the degeneracy is the fastest possible.

# 2   Main result

Consider a closed (i.e., connected, compact, without boundary) Riemannian manifold on which qudits are laid down. We do not consider microscopic fermions here. We use a positive constant $a$ throughout as the lattice spacing: we assume always that in a ball of radius $a$ there are $\mathcal{O}(1)$ dimensional degrees of freedom and the diameter of the system is $L$ that is much larger than the lattice spacing $a$. Here, the diameter of a manifold is the distance between two points, maximized over all possible pairs of points. Let $\Pi$ denote a subspace[1] of the full Hilbert space $\mathcal{H}$ of the qudits. We will use the same symbol $\Pi$ to denote the orthogonal projector onto the subspace $\Pi$. For two different operators $O_1$ and $O_2$ that preserve the subspace $\Pi$ (i.e., $[\Pi, O_1] = [\Pi, O_2] = 0$) it may happen that $(O_1 - O_2)\Pi = 0$, in which case we say $O_1$ and $O_2$ are *equivalent* on $\Pi$. In general there are many equivalent operators that induce a given linear transformation on $\Pi$.

We define that a set $A$ of qudits is *correctable against erasure* or simply *correctable* with respect to $\Pi$, if for every linear transformation on $\Pi$, there exists an operator supported on the complement of $A$ that induces the transformation.[2] That is, the complement of a cor-

---

[1]It is legitimate to think of $\Pi$ as the ground state subspace of some Hamiltonian, but our argument will have nothing to do with a Hamiltonian.

[2]An equivalent, perhaps better known condition for the correctability is that for all operators $O_A$ supported on $A$

rectable subset of qudits supports a complete set of operators for $\Pi$. So, even if a correctable set becomes inaccessible to a thought experimentalist, the state in $\Pi$ can be manipulated to reconstruct the whole system — the system is "corrected" from erasure. Now, we define that a subspace $\Pi$ has *homogeneous topological order* if any set $A$ of qudits whose distance $a$ neighborhood is contained in the union of some disjoint (topological) balls in the manifold, is always correctable. Colloquially speaking, with homogeneous topological order, any region that deformation-retracts to a discrete set of points has to be correctable.

We can now state our bound:

**Theorem 1.** *Suppose that our closed space manifold $M$ of dimension $d \geq 2$ has the metric normalized such that its diameter $L$ is 1, and that the local Hilbert space dimension of the degrees of freedom within any ball of radius $a$ is $\mathcal{O}(1)$. Suppose that $\Pi$ on the Hilbert space of qudits on $M$ has homogeneous topological order. Then*

$$\log \dim_{\mathbb{C}} \Pi \leq c\mu(L/a)^{d-2} \tag{1}$$

*for some constant $c$ that depends only on the isometry class of $M$ and $\mu$ that depends only on the the density of degrees of freedom.*

*In addition, if $M$ is the standard $d$-sphere, then $\dim_{\mathbb{C}} \Pi = 1$. If $d = 2$, then $c$ is the (demi)genus of $M$.*

The normalization of the metric such that $L = 1$ is not important to the result, but we mention it to disambiguate what the isometry class means; if the diameters are different, two spaces cannot be isometric. Note also that an isometry is automatically a homeomorphism.

Assumptions of similar flavor for two-dimensional systems were considered in [10] and it was concluded that $\dim_{\mathbb{C}} \Pi \leq \mathcal{O}(1)$. The proof of the theorem below will be an application of an idea by Bravyi, Poulin, and Terhal [11], augmented by a new topological argument to handle arbitrary closed Riemannian manifolds. The latter is our main technical contribution.

## 3 Generality

Before we present a formal proof of the theorem, it is important to understand the generality of the condition. First, a discrete gauge theory (or the toric code [12]) has homogeneous topological order for the following reason. In this model, any operator that commutes with the ground space projector $\Pi$ is a (co)homological cycle. The action of such an operator depends on the (co)homology class it represents, and there always exists a representative that goes around any given ball or a given collection of disjoint balls. Thus, we can always find a complete set of representatives for operators on $\Pi$ in the complement of the balls. This implies that the discrete gauge theory has homogeneous topological order.

Another way to see why any anyon model has to obey our homogeneous topological order condition is to consider an equivalent notion of correctability [10]. A set $A$ of qudits is correctable against erasure with respect to $\Pi$ if for arbitrary transformation on $\Pi$ by an operator supported on $A$ can be reversed by some transformation on the whole system; this has to be true intuitively, since the full subspace $\Pi$ can be accessed on the complement of $A$, on which the adversarial operation did not act. Now, imagine that $A$ is a disk. For an anyon model, an operator $O_A$ on $A$ would create some anyons from the ground state, but by locality of the Hamiltonian those anyons should be located near $A$. Since (a small neighborhood of) $A$ does not contain any topologically nontrivial loop, if the anyons are pushed towards the center of

---

there is a complex number $c(O_A)$ such that $\Pi O_A \Pi = c(O_A)\Pi$. This equation means that any operator on $A$ acts trivially on $\Pi$. This formulation is known as the Knill-Laflamme criterion [9].

*A*, then the anyons should fuse to vacuum with certainty. The overall procedure from the creation of the anyons by $O_A$ to the fusion into the vacuum, is happening near *A* and thus should not induce any nontrivial transformation on $\Pi$. That we push the anyons and fuse them to vacuum, amounts to a recovery operation. Even if *A* consists of several disks, the fusion can happen in the individual disks and overall we obtain a recovery operation.

This picture continues to hold in fracton models. For example, in the cubic code model [13], even though a single excitation is immobile, it can be pushed at the expense of creating others, and if a cluster of excitations is created by a local operator then the cluster always fuses into the vacuum. It can be shown [14] that indeed this leads to a recovery operation and the cubic code model has our homogeneous topological order. We believe that the homogeneous topological order condition is satisfied for all explicit fracton constructions to date. At least, all the cubic codes [13], the X-cube model, and the checkerboard model [15] have homogeneous topological order; see Section 3.1 below. Rigorous verification is anticipated for other models [6,7] in flat space or general manifolds [4,16].

Our setting assumes finite dimensional degrees of freedom, and hence does not immediately cover theories with $U(1)$ degrees of freedom [17, 18]. However, in these theories the degeneracy should be counted in units of $U(1)$ degrees of freedom, which requires some regularization.

We will handle situations where the recovery of a correctable set is not perfect; see the last section on approximate recovery. The same conclusion will hold under a relaxed, approximate setting.

## 3.1   Translation invariant exact code Hamiltonians

A translation-invariant exact code Hamiltonian $H = -\sum_j h_j$ [19] is an unfrustrated Hamiltonian on an infinite lattice with commuting terms $h_j$ each of which is a tensor product of Pauli matrices (Pauli operator) such that any *finitely* supported Pauli operator that commutes with every Hamiltonian term is a product of Hamiltonian terms up to a phase factor. Explicit examples are the cubic code models [13] and the X-cube and checkerboard model [15]. Here let us show that *the ground state subspace of such a Hamiltonian on any periodic finite lattice has the homogeneous topological order.* Our space manifold in this subsection is a *d*-torus $T^d$.

Since this Hamiltonian $H$ consists of commuting terms, all correlation functions decay abruptly to zero and hence the union of any two correctable regions is correctable whenever the two regions are so separated that no single term of the Hamiltonian can overlap the two regions simultaneously. (Proof: If $O_A$ and $O_{A'}$ are two operators on correctable regions *A* and *A'*, respectively, the Knill-Laflamme criterion [9] reads $\Pi O_A \Pi = c(O_A)\Pi$ and $\Pi O_{A'} \Pi = c(O_{A'})\Pi$. By pulling out local ground state projectors $\pi_j = \frac{1}{2}(I + h_j)$ from $\Pi = \pi_j \Pi$, we see that $\Pi O_A O_{A'} \Pi = \Pi O_A \Pi O_{A'} \Pi = c(O_A)c(O_{A'})\Pi$. By linearity, the Knill-Laflamme criterion is satisfied for all operators on *A* union *B*.)

Hence, it remains to show that the homogeneous topological order condition is obeyed over a region *A* (a set of qudits) of arbitrary size such that its *a*-neighborhood can be covered by a topological ball. That is, we have to show that *A* is always correctable. We choose the microscopic length *a* to be a sufficiently large constant (5 is enough) multiple of the interaction range of *H*. Here the interaction range means the minimum diameter of a ball that can cover the support of every term $h_a$ of *H*.

We check the Knill-Laflamme condition: for any operator $O_A$ on *A*, there exists a complex number $c(O_A)$ such that $\Pi O_A \Pi = c(O_A)\Pi$. First, as usual, we reduce the problem where $O_A$ is a tensor product of Pauli matrices. Indeed, if Knill-Laflamme condition is obeyed for every Pauli operator on *A*, then for any $\mathbb{C}$-linear combination $O_A = \sum_k \alpha_k P_k$ of Pauli operators on *A* we see $c(O_A) = \sum_k \alpha_k c(P_k)$. Now, if a Pauli operator $P_A$ on *A* does not commute with any Hamiltonian

term $h_j$, we know $P_A h_j = -h_j P_A$,[3] implying that $\Pi P_A \Pi = \Pi P_A h_j \Pi = -\Pi h_j P_A \Pi = -\Pi P_A \Pi = 0$, and the Knill-Laflamme condition is obeyed. So, the problem is further reduced to the case where the operator $O_A = P_A$ is a Pauli operator and commutes with every Hamiltonian term.

Let $B$ be a topological ball that contains the $a$-neighborhood of $A$.[4] Fix an arbitrary point of $B$ to consider the lift $\tilde{B}$ of $B$ into the covering space $\mathbb{R}^d$ of $T^d$. Our periodic lattice in $T^d$ is covered by an infinite lattice in $\mathbb{R}^d$. The lift $\tilde{B} \subset \mathbb{R}^d$ is a bounded topological ball, and the Pauli operator $P_A$ is lifted uniquely to a finitely supported Pauli operator $\tilde{P}$ on $\tilde{B}$. Since $a$ is much larger than the interaction range, the commutativity of $P_A$ with Hamiltonian terms implies that $\tilde{P}$ also commutes with every Hamiltonian term. By assumption on *exact* code Hamiltonians, it follows that $\tilde{P}$ is a product of some Hamiltonian terms on $\mathbb{R}^d$ up to a phase factor. But, a Hamiltonian term on $\mathbb{R}^d$ maps under the covering map to a Hamiltonian term on $T^d$. Therefore, $P_A$ is a product of Hamiltonian terms up to a phase factor, say $\eta$. Since $H$ is unfrustrated, i.e., each $h_j$ takes eigenvalue $+1$ on $\Pi$, we conclude that $P_A \Pi = \eta \Pi$. The Knill-Laflamme condition is obeyed with $c(P_A) = \eta$.

It requires calculation to check if a given Hamiltonian is an exact code Hamiltonian. This can be done by a polynomial method of [19], that is used for the X-cube and checkerboard model in [15], or by a more elementary method of [13] that is used for the cubic code models.

## 4 Relation to other notions

Our notion of homogeneous topological order is broader than the so-called "liquid" topological order [20, 21]. This is a notion given to a system-size-indexed family of many-body states which are interrelated by locality preserving unitaries with supply of ancilla qudits in a fixed state (entanglement RG). This condition is not satisfied by fracton phases [16, 22, 23]. On the other hand, our homogeneity allows us to consider a single finite system, rather than a family, as long as there is a clear hierarchy in the length scales $a \ll L$.

Unlike a mathematical topological quantum field theory as a functor from topological spaces with bordisms to vector spaces with linear maps [24], our notion does not aim to coherently bundle Hamiltonians on various manifolds. The notion is a finitary condition on a subspace of a Hilbert space with a locality structure. It is however robust under locality preserving unitaries: given a system of qudits, if a subspace $\Pi$ is homogeneously topologically ordered, then for any locality preserving unitary $U$ the transformed subspace $U\Pi U^\dagger$ is also homogeneously topologically ordered with a slightly increased lattice length scale $a$ by which one takes the neighborhood of a correctable set.

On a technical side, it is necessary for a sensible definition that we consider the $a$-neighborhood of a set $A$. Since we are considering a finite, discrete set of qudits sitting on a manifold, any set of qudits is covered by a disjoint union of (tiny) balls. So, without taking the $a$-neighborhood, our notion of homogeneous topological order would be vacuous. This technicality aligns well with the anyon-pushing intuition above, as the anyons are just near the operator they are inserted by, not necessarily right on top of the operator.

Bravyi, Hastings, and Michalakis [25] impose a condition that the local reduced density matrix of a ball-like region $A$ be completely determined by Hamiltonian terms that touch $A$, as long as $A$ has diameter $\mathcal{O}(L^\gamma)$ for some fixed $\gamma > 0$. Assuming this, they prove perturbation stability of the energy gap. If we identify our microscopic length scale $a$ with the interaction range of a Hamiltonian and replace their size restriction with our topological restriction on $A$,

---

[3]This assumes that the underlying degrees of freedom are qubits, but generalization is straightforward.

[4]In this argument it is possible that, for example, $A$ consists of sites near a "long line" $\{(t, 10t) \in \mathbb{R}^2/\mathbb{Z}^2 \mid 0.01 \leq t \leq 0.99\}$ in a 2-torus. This long line can be covered by a topological 2-disk, which cannot be contained in a rectangle of linear dimensions less than 1.

then their condition becomes our homogeneous topological order condition. See the equivalence of different correctability criteria [10]. While the identification of $a$ with the interaction range is natural, our requirement of the indistinguishability of $\Pi$ up to length scale $\mathcal{O}(L)$ is stronger than theirs because their $\gamma$ may be smaller than 1. However, we do not know any translation invariant Hamiltonian where the indistinguishability of the ground states is satisfied for $\mathcal{O}(L^\gamma)$-sized operators for some positive $\gamma < 1$ but not for $\mathcal{O}(L)$-sized operators. That is, as far as we know, for every translation invariant model on a torus of linear size $L$ the ground state subspace admits either a local observable or no observable at all within any box of linear size $L/2$.

## 5  Nonexamples

If a subspace $\Pi$ of $\mathbb{C}$-dimension greater than 1 admits a local observable, say $Z$, (as does the ground state subspace of the classical Ising model), then our homogeneous topological order condition does not hold. The local operator $Z$ whose eigenspectrum decomposes $\Pi$ ($Z$ is a local order parameter) must not commute with some operator $O$ that acts within $\Pi$, i.e., $[O, \Pi] = 0$ and $[O, Z] \neq 0$. Then, this operator $O$ or any equivalent operator (that is *conjugate* to $Z$) must have overlapping support with $Z$. Put differently, there cannot exist any operator $O'$ that is equivalent to $O$ ($(O - O')\Pi = 0$) such that $O'$ avoids the support of $Z$. Hence, the support of $Z$ is not correctable, though it is covered by a small ball. Therefore, the absence of local observables for $\Pi$ is necessary for our homogeneous topological order.

However, the lack of local observables is not sufficient. There is a non-translation-invariant gapped Hamiltonian [26] in three spatial dimensions that has a perturbation-stable ground space but fails to satisfy our homogeneous topological order. This is a network of $\mathbb{Z}_2$-gauge theory blocks where each block has linear size $L^{2/3}$ and there are $L^{1/3}$ blocks along each of three spatial directions, "welded together" in a certain way. The overall system is embedded in a 3-torus of linear size $L$. At a scale where $L^{2/3}$ is a unit length, the system looks like an Ising model. This example breaks our homogeneity condition because a ball that contains a gauge theory block is still very small compared to the total system, but is not avoided by some operator on the ground space.

In a similar vein, it is rather trivial to break our homogeneity while keeping the ground space as an error correcting code and keeping the perturbation-stability of energy gap above the ground space. Namely, one can simply introduce intermediate length scale $\ell$ such that $a \ll \ell \ll L$ and juxtapose $(L/\ell)^d$ boxes in $d$ dimensions, each of which has homogeneous topological order and has linear size $\ell$.

These nonexamples illustrate that the length scale at which observables for $\Pi$ start to appear has to be truly macroscopic. We have evaded the inhomogeneity due to intermediate length scales by stating the condition in terms of the topology of the ambient space.

## 6  Proof of the theorem

Let us now prove the theorem rigorously. We will use the following result:

**Fact 2** (Cerf and Cleve [27]). *Let $A \sqcup B \sqcup C$ be a partition of the set of qudits, and $\Pi$ be an orthogonal projector on $\mathcal{H}_{ABC}$. If $A$ is correctable with respect to $\Pi$ and so is $B$, then $\dim_{\mathbb{C}} \Pi \leq \dim_{\mathbb{C}} \mathcal{H}_C$ where $\mathcal{H}_C$ is the Hilbert space of qudits in $C$.*

*Proof.* One of the equivalent conditions of the correctability is that no matter how a state $\rho^{ABCR}$ that commutes with $\Pi = \Pi_{ABC}$ is entangled between $ABC$ and a reference system $R$, the mutual

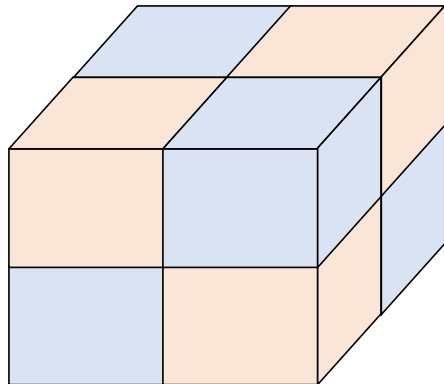

Figure 1: A cellulation of a $d$-torus with two colors ($d = 3$). The union of the cells of one color becomes a disjoint union of topological balls upon deletion of a small neighborhood of the $(d-2)$-skeleton.

information between a correctable region $A$ and $R$ is always zero: $S(\rho^A) + S(\rho^R) - S(\rho^{AR}) = 0$ where $S$ is the von Neumann entropy [27]. When the overall state on $ABCR$ is pure, the zero mutual information condition reads $S(\rho^A) + S(\rho^{ABC}) - S(\rho^{BC}) = 0$. Similarly for $B$, we have $S(\rho^B) + S(\rho^{ABC}) - S(\rho^{AC}) = 0$. The subadditivity of entropy implies $S(\rho^A) + S(\rho^B) + 2S(\rho^{ABC}) = S(\rho^{AC}) + S(\rho^{BC}) \leq S(\rho^A) + S(\rho^B) + 2S(\rho^C)$, or $S(\rho^{ABC}) \leq S(\rho^C)$ for any state in $\Pi$. Taking the maximally entangled state between $\Pi_{ABC}$ and $\mathcal{H}_R$, we have $S_{ABC} = \log \dim_{\mathbb{C}} \Pi$. It is always true that $S_C \leq \log \dim_{\mathbb{C}} \mathcal{H}_C$. $\qquad \square$

*Proof of Theorem 1.* Let us handle the simple case of a sphere. The northern and southern hemispheres are both balls. Hence, letting $A$ and $B$ be these hemispheres, respectively, and letting $C$ be empty in Fact 2, we see that $\dim_{\mathbb{C}} \Pi = 1$.

For $M$ with general topology, we consider a cellulation of $M$ with the following properties. This is to apply Fact 2 following [11].

1. Each $d$-cell is colored by either red or blue and is topologically an embedded ball.

2. If we delete the $10a$-neighborhood of the $(d-2)$-skeleton, the union of cells of the same color consists of topological balls separated by distance $> 2a$.

For a $d$-torus, a checkerboard cellulation as in Fig. 1 qualifies.

Given such a cellulation, let $C$ be the set of qubits within $10a$-neighborhood of the $(d-2)$-skeleton. The number of qudits in $C$ is $\mathcal{O}((1/a)^{d-2})$ where the hidden constant depends on the geometry of the cells, but not on $a$. This statement is most easily proved by considering a cover $\{U_\alpha\}_\alpha$ of $k$-skeleton $M^k = \bigcup_\alpha U_\alpha$ and appeal to the compactness of $M$. Since there are only a constant number (depending on the topology of $M$) of $k$-cells in $M^k$ we consider $M^k$ as if it consisted of a single smooth $k$-dimensional cell with metric inherited from $M$. We may choose each $U_\alpha$ to be a sufficiently small open ball in which every geodesic sphere of radius $r$ has volume $V_k r^k$ within a factor of 2, where $V_k = \pi^{k/2}/\Gamma(1 + \frac{k}{2})$ is the volume of the unit Euclidean $k$-ball. (In a curved space the volume of a radius $r$ ball has curvature-dependent higher order corrections. See [28] and references therein.) Since $M^k$ is compact, a finite subcover $\{U_{\alpha_1}, \ldots, U_{\alpha_n}\}$ covers $M^k$. Hence, for any $r$ that is smaller than the diameter of any $U_{\alpha_j}$ where $j = 1, \ldots, n$, the volume of any radius $r$ ball of $M^k$ may be computed as if it were a Euclidean ball up to a uniform constant. Hence, the $k$-skeleton $M^k$ has volume $\mathcal{O}((L/a)^k)$ in units of radius-$a$ dimension-$k$ balls. The $10a$-neighborhood of $M^k$ within $M = M^d$ has volume

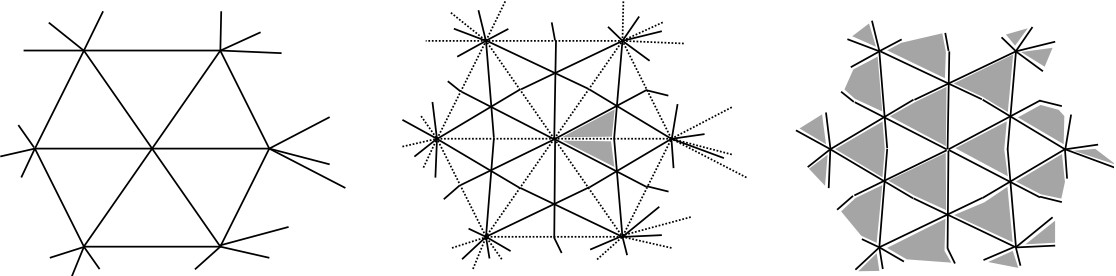

Figure 2: $d = 2$ (Left) A portion of the $(d-1)$-chain $\Delta'$. (Middle) The dashed $(d-1)$-simplices form $(\Delta')'$ of $M''$. The solid $(d-1)$-simplices form the chain $N = \Delta'' - (\Delta')'$. The shaded $d$-simplices are a pair of a $d$-simplex and its partner. (Right) The shaded $d$-cells form the chain $P$ whose boundary is $N$.

$\mathcal{O}((L/a)^k)$ in units of radius-$a$ dimension-$d$ balls, in each of which there are $\mathcal{O}(1)$ qudits by assumption.

Let $A$ be the set of all qudits in the red cells but not in $C$. By the second condition of our cellulation, the $a$-neighborhood of $A$ is covered by a disjoint union of balls. Since $\Pi$ has homogeneous topological order, $A$ is correctable. Similarly, let $B$ be the set of all qudits in the blue cells but not in $C$, and we see that $B$ is correctable. Now, Fact 2 implies the theorem.

It remains to show the existence of the desired cellulation for any given closed Riemannian manifold $M$. (We already have proved the existence if $M$ is a $d$-torus.) By Whitehead's theorem [29], $M$ that is a smooth manifold admits a triangulation. Let $M'$ be the barycentric subdivision of the triangulation, and $M''$ be the barycentric subdivision of $M'$. ($M''$ is the second barycentric subdivision of $M$.) Let $\Delta'$ be the homological $\mathbb{Z}_2$-chain that is the sum of all $(d-1)$-cells of $M'$. Likewise, let $\Delta''$ be the $\mathbb{Z}_2$-chain of all $(d-1)$-cells of $M''$. Under the barycentric subdivision $M' \mapsto M''$, every $k$-simplex $s$ of $M'$ can be thought of as the union of all $k$-simplices of $M''$ that are subsets of $s$; this association defines a unique map B from the $k$-chain group of $M'$ to the $k$-chain group of $M''$ with $\mathbb{Z}_2$ coefficients. Let $(\Delta')'$ be the image of $\Delta'$ under B, and define $N = \Delta'' - (\Delta')'$, a $(d-1)$-chain of $M''$. See the left and middle figures of Fig. 2.

Every $d$-simplex of $M''$ has a unique "partner" $d$-simplex that shares a $(d-1)$-face which is a nonzero summand in $(\Delta')'$. Let $Q = \{(s, \text{partner of } s)\}$ be the collection of all the pairs of a simplex $s$ and its partner.

By Whitney's theorem [30], $\Delta'$ represents the Poincaré dual of the first Stiefel-Whitney class $w_1$, and so does $\Delta''$. Since $\Delta''$ and $(\Delta')'$ are homologous, the chain $N$ is null-homologous; $N = \partial P$ for some $d$-chain $P$ of $M''$. See the right figure of Fig. 2. If $P$ has a $d$-simplex as a nonzero summand, then $P$ must have its partner as a nonzero summand as well, since the $(d-1)$-face between a simplex and its partner is absent in $N$. Therefore, we may identify $P$ with a subcollection $Q_{red} \subset Q$ and define $Q_{blue} = Q \setminus Q_{red}$.

The two subcollections $Q_{red}, Q_{blue}$ give a desired cellulation of the space as follows. We merge each simplex and its partner to form a $d$-cell. Under this merger, $Q_{red}$ is a collection of topological $d$-balls such that any two different balls do not share a $(d-1)$-face; if they did, $\partial P$ would not have that $(d-1)$-face as a nonzero summand. Hence, if we delete a small neighborhood of the $(d-2)$-skeleton of $M''$, $Q_{red}$ becomes a collection of separated balls. The same is true for $Q_{blue}$.

The construction of $Q_{red}$ and $Q_{blue}$ depends on the triangulation of $M$, not on $a$. We have completed the existence of the desired cellulation starting with any triangulation of $M$. This completes the proof of the theorem in the general case.

Let us prove the special case of $d = 2$. We treat the orientable and nonorientable cases

separately. An orientable surface is a connected sum $M = \#^g T^2$ of $g$ two-tori $T^2$. Fix a triangulation on $T^2$, and let us build a triangulation on $M$ by keeping all triangles on each $T^2$ except for one or two triangles. The dropped triangles are to glue the tori. Thus we obtain a triangulation of $M$ where each of the numbers of triangles, edges, and vertices is $\mathcal{O}(g)$, where the hidden constant depends only on the initial triangulation of $T^2$. The last statement remains true even if we subdivided the triangulation as above, albeit with an increased hidden constant in $\mathcal{O}(g)$. These simplices are not isometric and may have widely different area or length; they depend on the embedding of the simplices into the surface that has metric. We have shown that $\log \dim_{\mathbb{C}} \Pi$ is bounded by the density of degrees of freedom times the volume of the $a$-neighborhood of the 0-skeleton. This is again $\mathcal{O}(g)$, independent of the metric on the surface; this is a special property of $d = 2$. This completes the argument for orientable surfaces. A nonorientable surface is a connected sum of projective planes so a completely parallel argument shows that $\log \dim_{\mathbb{C}} \Pi$ is at most linear in the demigenus, independent of the metric. $\qquad\square$

If $M$ were orientable, then the first Stiefel-Whitney class $w_1$ vanishes, and we did not have to consider the second barycentric subdivision $M''$; $\Delta'$ would already be null-homologous and it would suffice to let $Q_{red}$ be the $d$-chain whose boundary is $\Delta'$.

One may consider a hyperbolic surface with a constant negative Gaussian curvature, modded by a suitable group action to obtain a sequence of compact surfaces with growing genus. Having a constant curvature means that this collection of surfaces have local patches that are isometric; however, this does not mean that two surfaces in this collection are globally isometric since they cannot even be homeomorphic.

## 7 Approximate recovery

Theorem 1 has assumed the perfect recovery from erasure errors. This is of course an idealization that does not hold generically. However, we note that a certain approximate correctability suffices for essentially the same degeneracy bound to hold.

We begin with a notion of approximate correctability [10]. We say that a subset $X$ of qudits is $\delta$-*avoided* with respect to $\Pi$ if for every unitary operator $U^{XX^c}$ that commutes with $\Pi$ there exists an operator $V^{X^c}$ supported on the complement $X^c$ of $X$ such that

$$
\begin{aligned}
\left\| V^{X^c} \right\| &\leq 1, \\
\left\| (U^{XX^c} - V^{X^c})\Pi \right\| &\leq \delta, \\
\left\| \Pi(U^{XX^c} - V^{X^c}) \right\| &\leq \delta.
\end{aligned}
\tag{2}
$$

If $\delta = 0$, this reduces to our earlier definition of the exact correctability because any linear transformation is a $\mathbb{C}$-linear combination of unitaries (actually at most four unitaries). Then, our *approximate* homogeneous topological order condition is defined by requiring $\delta$-avoidance for any subsystem whose "infinitesimal" neighborhood (enlargement by the lattice spacing $a$) is covered by a collection of disjoint topological balls. In other words, for a ground state subspace with approximate homogeneous topological order, one can always achieve any linear transformation by acting on any subsystem that circumvents ball-like regions at the cost of small error $\delta$ in operator norm. Note that we do not require $\delta$ to approach zero in a large system size limit; however, we expect that $\delta$ can depend on $a$ by which one takes the neighborhood of an avoided region.[5]

---

[5]A reader may wonder what if we started with an approximate Knill-Laflamme condition such as "$\|\Pi O \Pi - c(O)\Pi\| < \epsilon$." It appears that there is too large a Hilbert space dimension factor to guarantee the conclusion of Lemma 3.

With these definitions, Fact 2 generalizes as:

**Lemma 3.** *Let $A \sqcup B \sqcup C$ be a partition of the set of qudits, and $\Pi$ be an orthogonal projector on $\mathcal{H}_{ABC}$. If $A$ is $\delta$-avoided with respect to $\Pi$ and so is $B$, then*

$$(1 - 27\delta \log \tfrac{1}{\delta}) \log \dim_{\mathbb{C}} \Pi \leq \log \dim_{\mathbb{C}} \mathcal{H}_C. \tag{3}$$

*Therefore, with $\delta$ sufficiently small, $\log \dim_{\mathbb{C}} \Pi$ is at most proportional to the volume of $C$.*

We have shown above that for any system on a closed Riemannian manifold there is a partition such that each of $A$ and $B$ is a collection of disjoint topological balls and $C$ is (an "infinitesimal" $a$-neighborhood of) a codimension 2 subsystem. Hence, our approximate homogeneous topological order condition with $\delta$ small enough, implies that $\log \dim_{\mathbb{C}} \Pi$ may scale at best as the volume of a codimension 2 subsystem. This is a generalization of Theorem 1 to this approximate setting.

Elements of the proof below have appeared in [10] where various notions of approximate correctability are studied, so we will be brief and use comfortable, nonoptimal inequalities.

*Proof.* If a subset $X$ is $\delta$-avoided, then for an arbitrary state $\rho^{XX^cR}$ between $\Pi$ and an arbitrary external system $R$ (also called a reference system) the reduced density matrix $\rho^{XR}$ obeys [10, Thm. 8]

$$\tfrac{1}{2} \left\| \rho^{XR} - \omega^X \otimes \rho^R \right\|_{\mathrm{tr}} \leq 3\delta \tag{4}$$

for some fixed state $\omega^X$. Being close to a product state, $\rho^{XR}$ has small mutual information (assuming $\delta < \frac{1}{10}$) [10, App. F]:

$$I_\rho(X : R) \leq 27(\delta \log \tfrac{1}{\delta}) \log d_R, \tag{5}$$

where $I_\rho(X : R) = S(\rho^X) + S(\rho^R) - S(\rho^{XR}) \geq 0$, $S$ denotes the von Neumann entropy, and $d_R$ is the Hilbert space dimension of the subsystem $R$.

Suppose $d_R = \dim_{\mathbb{C}} \Pi$ and consider a maximally entangled state $\rho^{ABCR}$ between $\Pi$ and $R$, which satisfies for $X = A, B$

$$I_\rho(X : R) \leq 27(\delta \log \tfrac{1}{\delta}) S(\rho^R), \tag{6}$$

where $S(\rho^R) = \log \dim_{\mathbb{C}} \Pi$. Considering $I_\rho(A : R) + I_\rho(B : R)$ we see that

$$S(\rho^A) + S(\rho^B) + 2S(\rho^R) \leq 54(\delta \log \tfrac{1}{\delta}) S(\rho^R) + S(\rho^{AR}) + S(\rho^{BR}). \tag{7}$$

But the maximally entangled state is pure, so $S(\rho^{AR}) = S(\rho^{BC})$ and $S(\rho^{BR}) = S(\rho^{AC})$. Using subadditivity, Eq. (3) follows. $\qquad\square$

# Acknowledgments

I would like to thank Andrey Gromov and Zhenghan Wang for discussions.

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
