# Peer review of "A degeneracy bound for homogeneous topological order"

_SciPost Physics, doi:SciPost Phys. 10, 011 (2021)_

## Round 1 · Referee Report · Anonymous (Referee 1) · 2020-10-29

Strengths

1) The paper introduces a new notion of homogeneous topological order and clearly demonstrates its generality and usefulness.

2) The main theorem is a highly nontrivial result, and the paper makes a strong case that the required assumptions are relatively minor.

3) The proof of said theorem is concise and elegant.

4) The paper is clearly written and well-motivated.

Weaknesses

1) The paper provides relatively little intuition for what systems' ground state subspaces fail to have homogeneous topologically order.

Report

This work introduces a new, sharply defined notion called homogeneous topological order, and uses it to provide an elegant proof of an important bound on the ground state degeneracy of a wide variety of systems. It has long been believed that there is no fracton topological order in d=2, and this bound is a powerful and very general step towards that result. The bound also strongly constrains possible fracton-like topological orders in d>2. The fact that the homogeneous topological order condition can be used to concisely prove such a bound strongly suggests that it is a broadly useful condition and merits significant further investigation.

This is an excellent paper. It contains elegant and highly nontrivial new physics, is well-motivated, and is clearly written. I have a few relatively minor requested changes (see below), but I strongly recommend publication.

Requested changes

1) As written, it is not intuitively clear when one should expect the ground state subspace of a system to be a nonexamples, so it is unclear whether the given nonexamples are obvious ones or surprising ones. It would be helpful to elaborate on this. This might even involve mentioning some fairly trivial nonexamples, such as when \Pi contains locally distinguishable states, to emphasize the connection to prior notions of topological order.

2) If feasible, an appendix demonstrating how to prove that an example fracton model (say, X-cube) has homogeneous topological order would help the present paper be more self-contained.

3) A suggestion either for an additional comment in the paper (if the answer is straightforward) or followup work (if not): if a homogeneously topologically ordered subspace is used as a topological error-correcting code, to what extent can the technique used to prove the main theorem be used to restrict the form of logical operators?

---

## Round 1 · Referee Report · Anonymous (Referee 2) · 2020-11-17

Strengths

1- Extremely clear 2- Proves a useful result 3- The proof involves a nice idea

Weaknesses

1- Does not solve all the problems of physics and mathematics.

Report

The result proved in this paper -- that under reasonable assumptions, the log of the groundstate degeneracy of gapped topological phases grows as $L^{d-2}$ or slower in $d$ dimensions -- is not a great surprise to experts in the field (I have seen it used as a heuristic to decide if a solvable lattice model exhibits a non-topological degeneracy). But it is nice to understand why it is true, under clear assumptions. I expect that the notion of "homogeneous topological order" will have other uses. I also expect that the proof technique will be useful for other questions.

Requested changes

None.

---

## Round 2 · Author Response

I would like to thank the referees for their generous comments. One of the referees had specific questions, which I answer as follows.

1) The referee asked to include discussions for models with local order parameters. I restructured the section on nonexamples to discuss such models.

2) The referee asked to include proofs for simple specific fracton models that they have homogeneous topological order. I wrote a new subsection to give a full discussion for topologically ordered (in a conventional sense) translation invariant Pauli stabilizer codes.

3) The referee wondered to what extent the technique in the manuscript can constrain shape of logical operators. This is an important question and deserves deeper look.

3) A suggestion either for an additional comment in the paper (if the answer is straightforward) or followup work (if not): if a homogeneously topologically ordered subspace is used as a topological error-correcting code, to what extent can the technique used to prove the main theorem be used to restrict the form of logical operators?

--- This deserves deeper investigation.

---

## Round 2 · List of Changes

- Strengthened the statement of the theorem for d=2.
- Mentioned hyperbolic surface of constant curvature.
- Gave one more reference for the asymptotic volume formula of $a$-neighborhood of $(d-2)$-skeleton.
- Restructured the section on nonexamples, which now includes $\Pi$ with local observables.
- Wrote a new subsection 3.1 to show that translation invariant exact code Hamiltonians (a certain class of Pauli stabilizer code) have our homogeneous topological order.

---

## Editorial Decision

published